# Deep Neural Networks-Based Age Estimation of Cadavers Using CT Imaging of Vertebrae

**DOI:** 10.3390/ijerph20064806

**Published:** 2023-03-09

**Authors:** Hiroki Kondou, Rina Morohashi, Hiroaki Ichioka, Risa Bandou, Ryota Matsunari, Masataka Kawamoto, Nozomi Idota, Deng Ting, Satoko Kimura, Hiroshi Ikegaya

**Affiliations:** Department of Forensic Medicine, Graduate School of Medicine, Kyoto Prefectural University of Medicine, 465 Kajiicho, Kawaramachi-dori Hirokoji-agaru, Kamigyo-ku, Kyoto 602-8566, Japan

**Keywords:** forensic medicine, age estimation, postmortem computed tomography, deep neural network, artificial intelligence

## Abstract

Although age estimation upon death is important in the identification of unknown cadavers for forensic scientists, to the best of our knowledge, no study has examined the utility of deep neural network (DNN) models for age estimation among cadavers. We performed a postmortem computed tomography (CT) examination of 1000 and 500 male and female cadavers, respectively. These CT slices were converted into 3-dimensional images, and only the thoracolumbar region was extracted. Eighty percent of them were categorized as training datasets and the others as test datasets for both sexes. We fine-tuned the ResNet152 models using the training datasets. We conducted 4-fold cross-validation, and the mean absolute error (MAE) of the test datasets was calculated using the ensemble learning of four ResNet152 models. Consequently, the MAE of the male and female models was 7.25 and 7.16, respectively. Our study shows that DNN models can be useful tools in the field of forensic medicine.

## 1. Introduction

Identifying unknown cadavers is one of the most important tasks for forensic scientists. For that, forensic scientists have to consider several factors for example, the sex, human race, therapeutic remains, etc. Among them, age is one of the most important clues for identification. Several methods have been developed to estimate the age of cadavers, such as evaluating the cranial suture closure, measuring the Raman spectrometry of the skin, evaluating the teeth, and measuring the bone mineral density (BMD) [1,2,3,4]. However, employing usable methods in putrefied cadavers has been limited. Particularly, we have to rely on hard tissues, such as bones or teeth.

The shape of the vertebrae and vertebral column can change as people get older [5]. Moreover, osteophyte formation, a compression fracture caused by decreased BMD, and spinal curvature can be recognized visually. Few studies have focused on the shape of the vertebrae for age estimation, and the error value was approximately 10 years [6,7]. They categorized the shape of the vertebrae by the degree of osteophyte formation and estimated the age based on it. Although these studies show that evaluating the degree of osteophyte formation can help estimate age, we hypothesized that the accuracy of estimation can be improved by including other age-related factors. To the best of the author’s knowledge, there is a lack of well-established and evidenced methods required to estimate the age based on the appearance of whole thoracolumbar. Machine learning (ML) can be a useful tool for obtaining vertebral characteristics. Recently, a deep neural network (DNN) model (an ML algorithm that has shown good results in past studies) was used in the medical field [8,9,10]. By training the DNN models using images of the whole vertebrae and vertebral column, these models can consider the aforementioned factors and provide good results. Additionally, one study showed that DNN models could surpass human performance for visual recognition when using high-quality images [11]. Based on these facts, we hypothesized that good estimates could be achieved by applying DNN models to high-quality vertebrae and vertebral column images. To the best of our knowledge, no studies have evaluated the usefulness of DNN models in age estimation using vertebrae and the vertebral column. In this study, we used computed tomography (CT) to obtain and evaluate vertebral shapes. By evaluating the 3-dimensional images of vertebrae and vertebral column that were built using CT slice images, performing autopsies to evaluate the shape of the vertebrae and vertebral column was unnecessary. Additionally, the number of CT examinations for cadavers has recently increased. Thanks to this, we can obtain the CT images more easily than before. The purpose of CT examinations for cadavers is mainly to diagnose the cause of death. However, if age estimation can be performed using CT images, the versatility of CT examinations can also increase. Hence, we examined the usefulness of the DNN models for age estimation using CT images of the vertebrae and vertebral column. We established the target criteria as the error value within 10 years for the previous methods.

## 2. Materials and Methods

### 2.1. Image Collection

This was an observational, retrospective study. We performed a postmortem computed tomography (PMCT) examination of cadavers at our institute from February 2021 to October 2022. An 80-Multi Detector-row CT scanner (Aquilion Helios, Canon Medical Systems, Tochigi, Japan) was used for PMCT scanning with a 0.5 mm slice and peak voltage of 135 kV. All the cadavers were scanned from the head to the pelvis in the supine position. We converted these 2-dimensional CT slice images into 3-dimensional images using a workstation (Ziostation2, Ziosoft, Inc., Tokyo, Japan). For the 3-dimensional image construction, only regions with Hounsfield Unit (HU) values over 100 were used. Further, we extracted only the thoracic and lumbar vertebrae. This workstation could extract regions of interest by specifying the necessary or unnecessary regions based on the continuity of the HU values and the shape of the structures specified by the examiner. Finally, we adjusted the 3-dimensional image to the front view and exported it as a JPEG file (Figure 1a). In some cases, such as spinal curvature, it was difficult to define the front view. Hence, the front view was defined as the front view of the 10th thoracic vertebrae. Through this procedure, we obtained a 3-dimensional image of the thoracolumbar region. Although it would be ideal to estimate the age based on the shape of one vertebra, because not all vertebrae correspond to age-related changes, this would be difficult. For example, a situation wherein some vertebrae have osteophyte formation and others do not even in a person is considered. We guessed that several vertebrae were needed. Therefore, we focused only on the thoracic and lumbar vertebrae to ensure a sufficient volume. This will further help detect spinal curvature as an age-related change of the vertebral column. If we considered the vertebrae from the cervical to the lumbar vertebrae, the size of each vertebra would be small. This might make it difficult to evaluate the shape of each vertebra.

In general, putrefied gas is generated as the postmortem interval prolongs [12]. In these putrefied cases, building 3-dimensional images was almost impossible because most of the vertebrae were replaced by putrefied gas. One article suggested using 3-dimensional images to train DNN models in the field of forensic medicine [13]. However, we hypothesized that if we used the 3-dimensional images of the vertebrae and vertebral column to train the DNN models, then these models could detect the internal vertebral defects caused by the gas, which could be used to estimate age. This was true for the 2-dimensional images of the CT slices. We assumed that 2-dimensional images that were converted from 3-dimensional images could reduce the effects of the putrefied gas by ignoring the vertebrae inside the gas. Hence, we converted the 3-dimensional images into 2-dimensional ones. Furthermore, in some cases, such as those involving charred bodies or skeletal remains, several vertebrae were either not aligned or missing. Hence, we established the exclusion criteria as follows: (1) cases wherein all the thoracic and lumbar vertebrae were completely replaced by gas and the shape of the vertebrae could not be recognized (Figure 1b); (2) cases where the vertebrae from the 1st thoracic to the 5th lumber were not aligned continuously or the vertebrae were defected because of injury, excluding compression fracture (Figure 1c); and (3) cases where the age of the cadavers was less than 18 years. These images were evaluated based on a discussion of one forensic radiologist and two forensic scientists. Using the above procedure, we obtained 1000 and 500 male and female images, respectively. To simplify the evaluation of the decomposition rate of these cadavers, we applied the vertebral putrefaction (VP) grading system. This method helps evaluate the decomposition of vertebrae in four grades based on the CT slice images. VP Grade 0 implies that there is no putrefied gas in the vertebral body. VP Grade 1 indicates that the putrefied gas occupied less than 1/3 of the vertebral body. VP Grade 2 implies that the putrefied gas occupies at least 1/3 but no more than half of the vertebral body. Finally, VP Grade 3 implies that the gas occupies more than half of the vertebral body [14]. In this study, we estimated the VP grade from the 1st thoracic vertebrae to 5th lumbar vertebrae, and we set the highest VP grade as the VP grade of the cadaver.

This study was approved by our institutional review board (ERB-C-2140).

### 2.2. Prediction Model

We applied a convolutional neural network (CNN) in this study. This is a DNN architecture that is often used for image recognition. A CNN uses input images as a tensor, and this tensor is multiplied by several filters. Through this process, specific features were identified. Because there are several filters and a CNN comprises multiple layers, several features from the target images were extracted. CNNs demonstrate good recognition accuracy.

The accuracy of a CNN model is likely to improve with the addition of layers. However, stacking many layers can cause vanishing/exploding gradients and degradation problems. Different approaches, such as batch normalization, have been developed to solve these problems [15]. A residual Network (ResNet) is a CNN model that addresses such degradation problems [16]. This degradation problem is a phenomenon wherein the improvement of the training error is saturated earlier in the deep-layer models than in the shallow-layer models, which consequently worsens the results. The authors of the ResNet suggested that this phenomenon might be caused by weight optimization problems, which led to the development of skip connections in ResNet. They are the mechanisms that allow propagation between distant layers in a deep neural network by bypassing multiple layers in the middle of the network and skipping several layers to connect to the next layer. Due to this system, propagation was maintained in the deeper model, thereby improving accuracy.

In this study, we applied ResNet152 to an age prediction model using fine-tuning. Fine-tuning means using the weights of already pre-trained neural network models by partially training the weight. This is a method used often when the number of training images is limited. Because the number of parameters to be trained is not large, training of neural network would also be ideal for small datasets. During this process, pre-trained DNN models with good accuracy in one task were retrained using new datasets. Although the number of retrained layers must be considered, fine-tuning makes it possible to speed up the training and achieve good results, even when using small datasets. ResNet152 is composed of five large CNN blocks. In this study, two small CNN blocks on the output side of the fifth CNN block, which is composed of three small CNN blocks, were retrained.

### 2.3. Data Augmentation

Data augmentation was useful for small datasets. By adding a few conversions to the images, this method contributed to increasing the number of datasets (for example, zooming in or out, rotation, and horizontal flip). However, the degree of conversion must be comprehensible, and unrealistic conversion must be avoided. In this study, we prepared datasets using CT images. This implies that the size of the vertebrae and vertebral column depended on the range of the CT scans. Furthermore, although we defined the frontal view as the front view of the 10th thoracic vertebra, an error of a few degrees occurred. The horizontal flip is also a realistic conversion. For data augmentation, we added a horizontal flip, maximum zoom in/out of 10%, maximum top/down and right/left migration of 6.25%, and maximum rotation of −10 to 10°.

Furthermore, we added the GridMask data augmentation to the training dataset. This method masks the original image using small, equally spaced squares [17]. Robustness against noise or shielding was ensured by covering certain areas. In this study, we divided the height of the image by 10, and the values were used as the height and width of the 10 masked squares. The training image obtained after converting the data augmentation is shown in Figure 2.

### 2.4. Training and Evaluation of Model

We randomly divided the male and female datasets into 80 and 20% training and test datasets, respectively, to perform the holdout examination. To avoid overfitting, we performed 4-fold cross-validation using the training datasets. This method continuously divides the training datasets into four groups, while the ResNet model is trained using three groups and evaluated using the rest four times. The hyperparameter sets for this training are listed in Table 1. The hyperparameters refer to the learning conditions. The loss function calculated the discrepancy between the predicted values calculated by the models and true values. Optimizers are used to converge the discrepancy, and the learning rate indicates the degrees of parameters changed during learning optimization. DNN models involve processing a large number of datasets, such as images, which requires a huge amount of personal computer memory. Therefore, simultaneously training the model using all the training datasets could exhaust the computer memory. Hence, all datasets were divided into a few groups, and the models were trained using these small groups. The batch size was the number of data points in a specific group. Epochs refer to the number of times the models were trained.

After 4-fold cross-validation, we obtained four trained ResNet models. We performed predictions for the test datasets using each of the four models and calculated the final prediction value for the test datasets using ensemble learning. Ensemble learning refers to uniting the results of several models. In this study, the prediction value for the test datasets was the mean of the four predictions from the four models.

The mean absolute error (MAE) was used to evaluate the model. The MAE was calculated using the following formula:$$\mathrm{MAE}=\frac{1}{n}\;\overset{n}{\underset{i=1}{\sum }}|{\mathrm{Prediction}\;\mathrm{value}}_{i}-{\mathrm{True}\;\mathrm{value}}_{i}|$$
where *n* stands for the number of datasets.

### 2.5. Training Environment

We built and trained the ResNet models using PyTorch (version 1.12.1+cu113) and PyTorch Lightning (version 1.7.7) with Python (version 3.9.7) on an NVIDIA GeForce RTX 3090. The operating systems used were Windows 10 Pro (version 22H2), CUDA 11.5, and cuDNN 8.3.

## 3. Results

### 3.1. Characteristics of Cadavers

As aforementioned, 1000 and 500 male and female cadavers, respectively, were used in this study. The information is presented in Table 2. In this study, the postmortem interval was defined as the duration from the expected time of death to the time of PMCT examination. We estimated the time of death based on the cadavers’ situation before their deaths and information from the police. However, in some cases, we could not estimate the time of death owing to insufficient information.

### 3.2. Result of 4-Fold Cross-Validation and Prediction for Test Datasets

The results are shown in Table 3.

Each of the four models showed relatively similar results for the training datasets. Moreover, the MAEs for the training datasets were fairly similar to the MAE for the test datasets. As a result, generalization performance could be obtained.

## 4. Discussion

Several age estimation methods use images, such as X-rays, CTs, or MRIs [18,19,20]. However, most age estimation studies have adopted machine learning methods, particularly linear regression models. Although one study used DNN models for head MRI images, these images were obtained from people living in the hospital, not cadavers [21]. Recently, DNN models have been gradually applied to forensic tasks, such as sex identification using CT images [22]. A review discussed the effectiveness of DNN models when applied to postmortem imaging in several studies [23]. However, these studies examined the utility of the DNN models using images of living people. As shown in Figure 1, there are several differences between living people and cadavers. Hence, applying DNN models that were trained with the datasets of living people to those of cadavers is not desirable. Furthermore, the DNN models that should be applied to forensic cases must be trained using forensic datasets. To the best of our knowledge, no study has examined the usefulness of artificial intelligence (AI) in the field of forensic medicine. Using CT images of vertebrae and vertebral column, we showed that DNN models were useful tools for age estimation. Based on the results, the four models showed relatively similar accuracy, and overfitting was unlikely. Additionally, the performance of the models was considerable. To the best of our knowledge, there were no estimation models to predict the age upon death within an MAE of 10 years using images, including for putrefied adult cases. Although estimating the age upon death completely with our models was difficult, they made it possible to narrow the range of the assumed age. This is the first study to demonstrate the possibility of using DNN models in forensic medicine. The number of forensic scientists worldwide is relatively small. However, forensic scientists’ tasks cover several different topics, such as the diagnosis of the cause of death and the identification of unknown cadavers. These tasks typically require vast knowledge and experience. For example, experienced forensic scientists may be able to estimate the age correctly according to the entire appearance of the thoracolumbar. However, due to these evaluations, a significantly greater time would be required to train forensic scientists; furthermore, they lack objectivity and reproducibility. Hence, if well-trained DNN models are introduced, they can be helpful tools with objectivity and reproducibility, particularly for inexperienced forensic scientists.

This study had several limitations. First, the number of training datasets was insufficient, especially in females. The DNN models typically require numerous datasets to avoid over-fitting. Although, one study demonstrated that good accuracy could be obtained by applying the appropriate network structure and nature of the dataset even with a small dataset and our model showed satisfactory results, increasing the number of female training datasets as much as males can potentially further improve the prediction performance [24]. Second, the effect of putrefied gas must be considered. We did not include completely putrefied cases. Putrefied gas sometimes affects the vertebral shape, thereby making the extraction of the thoracic and lumbar vertebrae difficult. For example, the transverse process was often deficient when building 3-dimensional images and extracting them. Additionally, when a large amount of putrefied gas occupied most of the vertebral bodies, the vertebral bodies looked defective in the front view. Although we presumed that the effect of putrefied gas could be reduced by converting 3-dimensional images into 2-dimensional ones, the small defects caused by putrefied gas remained even in the 2-dimensional images we used in this study. To imitate this defect artificially, we introduced GridMask for data augmentation. As aforementioned, the DNN models required numerous datasets. Furthermore, forensic scientists have to predict the age upon death in mostly putrefied or defective cases. Therefore, we included the putrefied cases if the shape of the vertebrae and vertebral column could be recognized. However, we could not apply our models to completely putrefied cases. We trained our models only with the instances of the thoracolumbar aligned completely. In some cases, such as with skeletal remains, the vertebrae were scattered, thereby making it difficult for our models to predict the age. Third, although it is an inevitable limitation in the studies related to forensic medicine, our datasets might be biased. Adjusting the background of cadavers could make the number of our datasets even smaller. In the first place, it is almost impossible to obtain all the information regarding the cadavers. Therefore, even if we standardize the dataset with the information that we obtained, it is possible that some background information related to cadavers could not be considered. However, because most forensic cases that we would aim to identify lack information such as past history, postmortem interval, and cause of death, building DNN models with standardized data would not be ideal as there would be an uncertainty regarding the DNN models that are trained with the standardized dataset to unknown background cases. In contrast, one study demonstrated that if the machine learning models are trained well, they can generalize across various conditions and bias will not necessarily matter [25]. Based on this evidence, adjusting and standardizing the dataset might not be necessary as our training did not fail. Finally, we could not establish the control to compare the performance of our model because there were no alternative methods that we could apply to our cadavers. In most cases, only postmortem CT examinations were performed. In other words, CT images were the only form of information available for use. Although there are several methods of predicting the age of cadavers with CT images, to the best of our knowledge, there are no methods that could be applied to putrefied cases. However, this implies that our methods could deal with the putrefied cases in that previous methods could not be applied. We hope this is one of the advantages of our model.

Future research could focus on developing methods using DNN models for completely putrefied or defective cases and even for one vertebra with an increased volume of data.

## 5. Conclusions

This study demonstrated the ability of the DNN models to estimate the age of cadavers even in putrefied cases. It is known that estimating the age of putrefied cases is difficult. However, newer approaches such as artificial intelligence for age assessment could be ideal for such cases.

## Figures and Tables

**Figure 1 ijerph-20-04806-f001:**
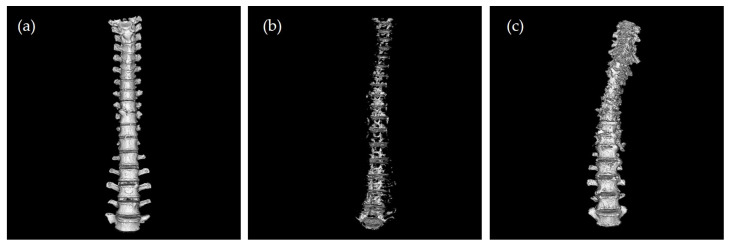
Examples of images that were included and excluded. (**a**) Included image: There was no defective or putrefied gas in the vertebrae body. (**b**) Excluded image: Most of the vertebrae bodies were replaced by putrefied gas and the shape of the vertebrae was unclear. (**c**) Excluded image: This is the case of death after burning. Although the lumbers remained, most of the thoracic vertebrae were defective.

**Figure 2 ijerph-20-04806-f002:**
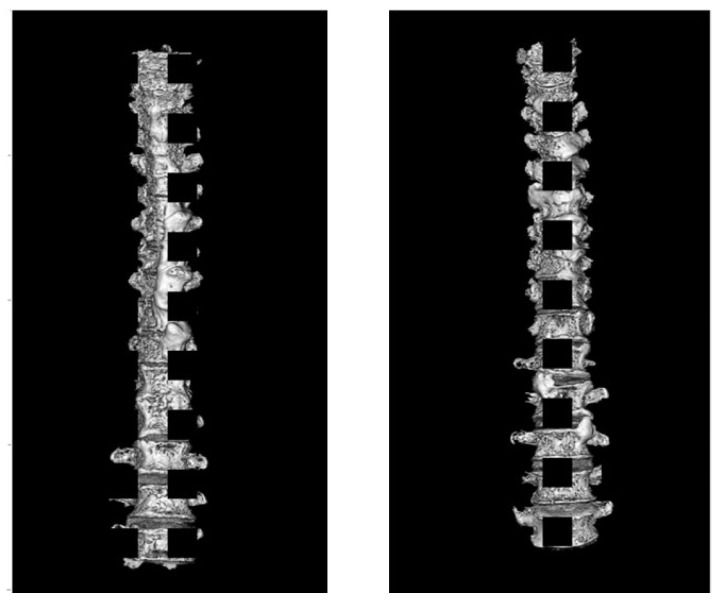
Images of converted augmentation. Scaling, rotation, horizontal flip, top/down and right/left migration, and GridMask were performed.

**Table 1 ijerph-20-04806-t001:** Hyperparameters in the study.

Size of the Input Image	(1024, 512, 3)
Loss function	L1Loss
Batch size	32
Learning rate	0.00001
Optimizer	AdamW
Epochs	3000

**Table 2 ijerph-20-04806-t002:** Characteristics of cadavers.

		Male	Female
Median age (interquartile range)		69 (58–76)	76 (64–84)
Minimum age		20	19
Maximum age		97	101
Postmortem interval range		0.5–301 days	0.5–240 days
The number of each VP Grades (percentages)	VP Grade 0	470 (47.0)	291 (58.2)
	VP Grade 1	194 (19.4)	92 (18.4)
	VP Grade 2	112 (11.2)	31 (6.2)
	VP Grade 3	224 (22.4)	86 (17.2)
The number of cadavers		1000	500

VP: Vertebral Putrefaction.

**Table 3 ijerph-20-04806-t003:** Results of 4-fold cross-validation.

		Male	Female
MAE for training datasets	Model 1	7.25	6.08
	Model 2	6.86	6.52
	Model 3	7.1	7.11
	Model 4	6.91	5.94
Ensemble of MAE for test datasets		7.25	7.16

## Data Availability

The datasets generated during and/or analyzed during the current study are available from the corresponding author on reasonable request due to opt-out.

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
