# Peer review of "Deep Neural Networks-Based Age Estimation of Cadavers Using CT Imaging of Vertebrae"

_ijerph, 2023, doi:10.3390/ijerph20064806_

Round 1

Reviewer 1 Report

The work is interesting, but contains some errors of method.

The first is champion selection. Since this is the first contribution of the use of artificial intelligence in the interpretation of the CT of the vertebrae to determine the age of the cadaver, it would have been desirable that the test was conducted on "test" cadavers, in which the age, the period of death and all other conditions were standardized. Using a very varied series, even if inserting exclusion criteria, does not allow to deduce which are the true results in relation to the knowledge of the phenomenon. This affects the scientific correctness of the results.

The second is constituted by the fact that the sample used for the construction of the algorithm is too small, especially for females. This method element cannot be corrected on the basis of the results obtained, which according to the authors are good. This is a scientifically incorrect methodology. The sample must be increse.

The third is represented by the fact that approval by an ethics committee is essential, and not by a committee that judges on the scientific validity of the project, as in the specific case. A reasoned opinion of an Ethics Committee on research is required.

For publication it is essential that the authors correct the three elements reported.

Author Response

We are grateful for the opportunity to resubmit the revised draft of our manuscript titled, “Deep neural-networks based age estimation of cadavers using CT imaging of vertebrae” to International Journal of Environmental Research and Public Health. The manuscript ID is ijerph-2193961.

We appreciate the time and effort you and the reviewers have invested in providing insightful feedback on ways to strengthen our manuscript. We have incorporated several changes based on these suggestions. We hope that the revisions move this manuscript closer to publication in International Journal of Environmental Research and Public Health.

We have rechecked the manuscript and made the necessary changes in accordance with the reviewers’ suggestions. We have provided point-by-point responses to the questions and comments raised below. We hope that our responses and the corresponding changes made in the revised manuscript (shown in the “Track Changes” function) satisfactorily address all the issues and concerns raised.

Thank you for your consideration. We look forward to hearing from you.

Response to reviewers

Reviewer 1: The first is champion selection. Since this is the first contribution of the use of artificial intelligence in the interpretation of the CT of the vertebrae to determine the age of the cadaver, it would have been desirable that the test was conducted on "test" cadavers, in which the age, the period of death and all other conditions were standardized. Using a very varied series, even if inserting exclusion criteria, does not allow to deduce which are the true results in relation to the knowledge of the phenomenon. This affects the scientific correctness of the results.

Our response: We thank the reviewer for the thorough and insightful feedback on our manuscript. As highlighted by the reviewer, the champion selection was required. However, there are several reasons why we could not prepare these data. First, in most forensic cases, it is almost impossible to obtain ALL information regarding the cadavers, such as occupational history, family history, and medical history. Even if we standardize and adjust the dataset by using the information that we obtained, it is possible that some background information related to cadavers that we were not aware of could not be considered. This means that adjusting and standardizing the dataset completely is almost impossible. This limitation is not just associated with our study, but with ALL forensic studies. Moreover, even if the datasets could be standardized and adjusted, and the usefulness of the artificial intelligence model is confirmed through these datasets, it may still not be clear whether the use of artificial intelligence to train the adjusted data could help estimate the age of cadavers with unknown background cases. In most forensic cases that we aim to identify, most information, such as past history, postmortem interval, and cause of death, is unknown. In other words, forensic scientists can almost never obtain useful information regarding the cadavers to be identified. Therefore, we should examine the usefulness of artificial intelligence not by the adjusted and ideal data, but by unadjusted and real data. If we conclude the use of the artificial intelligence model based on only the training data, the scientific correctness of the results could not be certified, as indicated by Reviewer 1. To ensure that this does not happen, the usefulness of our artificial intelligence model must be evaluated with an unknown test dataset (Therefore, the test dataset must not be used during the training process). Because this is a basic and typical Machine Learning process, we also followed the same. Second, to obtain the champion selection, a large volume of data will be required. In this study, we used all of the data that we could prepare. Standardizing or adjusting our datasets will reduce the number of datasets, even though it is already small. Thus, we established an artificial intelligence model by primarily using the data without standardizing.

We admit that this is the limitation of the application. However, we believe, that this concept corresponds to the theme of the Special Issue of your journal, “Artificial Intelligence as a Tool for Forensic Medicine: Future Challenges, Limits of Application and Critical Issues.”

We added this explanation in the “Discussion” part.

Reviewer 1: The second is constituted by the fact that the sample used for the construction of the algorithm is too small, especially for females. This method element cannot be corrected on the basis of the results obtained, which according to the authors are good. This is a scientifically incorrect methodology. The sample must be increse.

Our response: We thank the reviewer for the insightful suggestion. As highlighted by the reviewer, the higher the volume of available data, the higher the improvement in accuracy in most cases. However, the reason DNN models require a numerous number of datasets is to avoid over-fitting. In most medical image studies, it is well-known that the volume of data that researchers can obtain is not large. For these cases, methods such as fine-tuning or data augmentation are developed. The usefulness of those methods for small data cases has already been shown. Further, D'souza et al (doi:10.1038/s41598-020-57866-2) demonstrated that the use of an appropriate network structure and nature of the dataset even with the small dataset could yield good accuracy. They obtained this result by using small datasets (100, 500, and 1000 MNIST datasets). We used an appropriate network structure (ResNet152, which is a popular state-of-the-art model) and appropriate images (CT slices were 0.5 mm). As a result, on the basis of the gap between the result of validation data and test data, we believe that over-fitting did not occur and our training did not fail. We added this explanation in the “discussion” section.

Reviewer 1: The third is represented by the fact that approval by an ethics committee is essential, and not by a committee that judges on the scientific validity of the project, as in the specific case. A reasoned opinion of an Ethics Committee on research is required.

Our response: We thank the reviewer for the insightful suggestion. This study was approved by our institutional review board (ERB-C-2140).

Reviewer 2 Report

Age estimation of cadavers using CT imaging of vertebral column death is important in forensic investigations of unknown remains. However, this method in form presented here (DNNs and whole-column images) is not yet established.  The main objective of this study was to develop an age estimation technique from vertebral column CT-s processed and operated by artificial intelligence. And, if we disregard the modest size of a training sample, Kondou et al were successful - I appreciate the idea, but I have many reservations –

-        Please, distinguish “vertebrae” from “vertebral column”

-        In the 2014th retrospective study of Tan et al., CT had a sensitivity of 0.875 [0.719-0.950, 95% CI] and a specificity of 1.000 [0.930-1.000, 95% CI] in detecting all cervical spine injuries compared to MRI.  – so, I think it is not actually a top method to use. What’s even worse, authors didn’t bother at all to explain this. Neither to mention this. I miss some rationale foe this study, definitely – this idea lacks specificity – what does vertebral CT tell you? Anyway, this “mediocre sensitivity” is most likely the reason why “…only the thoracolumbar region was extracted…” – and I support it, only it needs an exolanation.

-        What do you mean “shape of the vertebrae” – this, as it reads now. Suggests shape of the single bone. If so, please – be specific. I guess, though, that author actually memt “the shape of vertebral column” – so, be specific.

-        The above was “recognized visually” – always by the same observer?

-        Better define” fine-tuning models”

-        Please, specify the type of NN used

-        If CNN is used - a rough rule of thumb is to train a CNN algorithm with a data set larger than 5,000 samples for effective generalization of the problem – this, unassuming, size is more for a pilot study -therefore – if I may suggest – pursue with publishing a pilot, and connect with other researchers in the field – to increase the sample size

Author Response

We are grateful for the opportunity to resubmit the revised draft of our manuscript titled, “Deep neural-networks based age estimation of cadavers using CT imaging of vertebrae” to International Journal of Environmental Research and Public Health. The manuscript ID is ijerph-2193961.

We appreciate the time and effort you and the reviewers have invested in providing insightful feedback on ways to strengthen our manuscript. We have incorporated several changes based on these suggestions. We hope that the revisions move this manuscript closer to publication in International Journal of Environmental Research and Public Health.

We have rechecked the manuscript and made the necessary changes in accordance with the reviewers’ suggestions. We have provided point-by-point responses to the questions and comments raised below. We hope that our responses and the corresponding changes made in the revised manuscript (shown in the “Track Changes” function) satisfactorily address all the issues and concerns raised.

Thank you for your consideration. We look forward to hearing from you.

Response to reviewers

Reviewer 2: Please, distinguish “vertebrae” from “vertebral column”

Our response: We apologized for the confusion in the previous manuscript. We supposed that our DNN models considered the shape of vertebrae such as osteophyte formation, compression fracture, and the shape of the vertebral column like spinal curvature. We have modified the highlighted sentence.

Reviewer 2: In the 2014th retrospective study of Tan et al., CT had a sensitivity of 0.875 [0.719-0.950, 95% CI] and a specificity of 1.000 [0.930-1.000, 95% CI] in detecting all cervical spine injuries compared to MRI.  – so, I think it is not actually a top method to use. What’s even worse, authors didn’t bother at all to explain this. Neither to mention this. I miss some rationale foe this study, definitely – this idea lacks specificity – what does vertebral CT tell you? Anyway, this “mediocre sensitivity” is most likely the reason why “…only the thoracolumbar region was extracted…” – and I support it, only it needs an exolanation.

Our response: We thank the reviewer for the insightful suggestion. As highlighted by the reviewer, the MRI is superior to CT in a similar manner as it is to the evaluation of cervical spine injuries. However, the main focus is to estimate the age. Thus, maximum information related to the cervical and sacrum is required to build a good model. When the MRI is used, it is difficult to include the shape from the cervical to the sacrum due to the range that MRI could scan. However, the CT could scan from the cervical to the sacrum easily. On the other hand, if the shape from the cervical to the sacrum is included in one image, the size of each vertebra may be small. This might make it difficult to evaluate the shape. Thus, we prepared the training image from the thoracic to the lumber. We added the relevant explanation in this article.

Reviewer 2: What do you mean “shape of the vertebrae” – this, as it reads now. Suggests shape of the single bone. If so, please – be specific. I guess, though, that author actually memt “the shape of vertebral column” – so, be specific.

Our response: We thank the reviewer for the insightful suggestion. We assumed that our DNN models considered the shape of vertebrae such as osteophyte formation, compression fracture, in addition to the shape of the vertebral column such as the spinal curvature. We modified the highlighted sentence in the revised manuscript.

Reviewer 2: The above was “recognized visually” – always by the same observer?

Our response: We evaluated the images through the discussion of one forensic radiologist and two forensic scientists. We added this explanation in the “2. Materials and Methods”

Reviewer 2: Better define” fine-tuning models”

Our response: Based on the suggestion of the reviewer, we added the explanations regarding “fine-tuning.”

Reviewer 2: Please, specify the type of NN used

Our response: We used a convolutional neural network (CNN) in this study. We added relevant explanation in “2. Materials and Methods.”

Reviewer 2: If CNN is used - a rough rule of thumb is to train a CNN algorithm with a data set larger than 5,000 samples for effective generalization of the problem – this, unassuming, size is more for a pilot study -therefore – if I may suggest – pursue with publishing a pilot, and connect with other researchers in the field – to increase the sample size

Our response: We thank the reviewer for the insightful suggestion. As highlighted by the reviewer, the higher the volume of available data, the higher the improvement in accuracy in most cases. However, the reason DNN models require a numerous number of datasets is to avoid over-fitting. In most medical image studies, it is well-known that the volume of data that researchers can obtain is not large. For these cases, methods such as fine-tuning or data augmentation are developed. The usefulness of those methods for small data cases has already been shown. Further, D'souza et al (doi:10.1038/s41598-020-57866-2) demonstrated that the use of an appropriate network structure and nature of the dataset even with the small dataset could yield good accuracy. They obtained this result by using small datasets (100, 500, and 1000 MNIST datasets). We used an appropriate network structure (ResNet152, which is a popular state-of-the-art model) and appropriate images (CT slices were 0.5 mm). As a result, on the basis of the gap between the result of validation data and test data, we believe that over-fitting did not occur and our training did not fail. We added this explanation in the “discussion” section. However, as stated by the reviewer, increasing the number of datasets could help improve the performance of DNN models. Thus, we have been gathering the images. We added this explanation in the “discussion”. (Certainly, we considered publishing this study as a pilot study, but article types of IJERPH are simply articles, reviews, and case reports.)

Round 2

Reviewer 1 Report

The author on the first observation of reviewer wrote: “it is possible that some background information related to cadavers that we were not aware of could not be considered. This means that adjusting and standardizing the dataset completely is almost impossible. This limitation is not just associated with our study, but with ALL forensic studies.

The sentence is incorrect. Forensic sciences are sciences in all respects and share the same scientific method with the other sciences. the rigor of the method, especially in scientific works submitted to international journals, must be the same.

A reasoned opinion of an Ethics Committee on research is required. The approvation of a institutional review board is non sufficient.

Author Response

Response to reviewers

Reviewer 1: The author on the first observation of reviewer wrote: “it is possible that some background information related to cadavers that we were not aware of could not be considered. This means that adjusting and standardizing the dataset completely is almost impossible. This limitation is not just associated with our study, but with ALL forensic studies.

The sentence is incorrect. Forensic sciences are sciences in all respects and share the same scientific method with the other sciences. the rigor of the method, especially in scientific works submitted to international journals, must be the same.

The first is champion selection. Since this is the first contribution of the use of artificial intelligence in the interpretation of the CT of the vertebrae to determine the age of the cadaver, it would have been desirable that the test was conducted on "test" cadavers, in which the age, the period of death and all other conditions were standardized. Using a very varied series, even if inserting exclusion criteria, does not allow to deduce which are the true results in relation to the knowledge of the phenomenon. This affects the scientific correctness of the results. (In the Round 1 revision)

Our response: Thank you for your thorough and insightful feedback on our manuscript. It is important to make sure the data in the test set is appropriate. Generally, a test dataset to evaluate machine learning models should follow the same probability distribution as the training dataset. Michael (arXiv:2108.02497) noted that “it should be representative of the wider population”. Andre et al. (https://doi.org/10.1038/s41379-022-01147-y) also noted that “test datasets must be diversified”. We agree with your assessment of “Forensic sciences are sciences in all respects and share the same scientific method with the other sciences.”. Standardizing all conditions of “test” cadavers can result in a loss of diversity and a different probability distribution from the training dataset. Therefore, we shared a well-known method, splitting the entire dataset randomly, which is one of the methods to mitigate the sampling bias. However, Jing et al. (arXiv:2009.13447) noted that “it is often hard or impossible to eliminate the sampling bias due to historical and operational issues.”. Wang et al. (doi: 10.1073/pnas.2211613120) also noted that “potential biases and poor generalization across genders, age distributions, races and ethnicities, hospitals, and data acquisition equipment and protocols” are problems when applying machine learning to medicine. However, they demonstrated that “when properly trained, machine learning models can generalize well across diverse conditions and do not necessarily suffer from bias.”. Based on this evidence, we believe that adjusting and standardizing the dataset is not necessary because our training did not fail. We have added an explanation to the “Discussion” section accordingly.

Reviewer 1: A reasoned opinion of an Ethics Committee on research is required. The approvation of a institutional review board is non sufficient.

Our response: Thank you for this insightful suggestion. At our institution, the institutional review board also considers ethical judgment. If an ethically inappropriate research proposal is planned, the institutional review board will not approve it. This study was approved by our institutional review board (ERB-C-2140), where ERB is the abbreviation for “Ethical Review Board.” If the certificate of approval is required, we can submit it.

Reviewer 2 Report

Tha you to the authors. This submission seems much improved, and I'm totally optimistic regarding this.

Author Response

Response to reviewers

Reviewer 2: Tha you to the authors. This submission seems much improved, and I'm totally optimistic regarding this.

Our response: Thank you for the appropriate review.